# Determinants of Government Expenditures with Health Insurance Beneficiaries in the Brazilian Health System

**DOI:** 10.3390/healthcare12232335

**Published:** 2024-11-22

**Authors:** Leonardo Moreira, João Vitor Marques Teodoro de Lima, Murilo Mazzotti Silvestrini, Flavia Mori Sarti

**Affiliations:** 1School of Arts, Sciences and Humanities, University of São Paulo, São Paulo 03828-000, Brazil; 2Institute of Economics, State University of Campinas, Campinas 13083-857, Brazil

**Keywords:** public healthcare facilities, ICD-10 diagnosis, financial burden, health system management, healthcare expenditures, high complexity healthcare

## Abstract

**Background/Objectives**: The Brazilian health system provides healthcare financed by the public and private sector, being the first designed to encompass universal healthcare coverage delivered to the population without charge to patients (Sistema Único de Saúde, SUS), whilst the second refers to healthcare coverage delivered for individuals with the capacity to pay for assistance through health insurance or out-of-pocket disbursements. Health insurance companies with beneficiaries receiving publicly financed healthcare from the SUS are required to provide the reimbursement of healthcare expenditures to the government, considering that the health insurance beneficiaries obtain deductions of income taxes designed to fund the SUS. Therefore, the study investigated patterns of healthcare utilization and public expenditure due to the use of public healthcare by beneficiaries of health insurance between 2003 and 2019. **Methods**: Datasets including annual information on healthcare utilization by beneficiaries of health insurance from the National Agency of Supplementary Health (Agência Nacional de Saúde Suplementar, ANS) were organized into a single database to allow for the identification of patterns of interest to inform public policies of health. The empirical strategy adopted included the estimation of regression models and agglomerative hierarchical cluster analysis to identify factors associated with public sector expenditure. **Results**: The regression results indicated lower expenditure with female patients, particularly children and adolescents under 20 years old, receiving treatment in public sector facilities linked to the federal government. The cluster analysis showed five types of health insurance beneficiaries with a higher level of healthcare utilization, being three clusters referring to medium complexity procedures with lower public expenditures, and two clusters with higher public expenditures, one cluster that refers to high complexity procedures, and one cluster referring to health insurance schemes without hospitalization. **Conclusions**: The findings of the study highlight the existence of patterns of healthcare utilization by health insurance beneficiaries that may compromise the sustainability of public funding within the Brazilian health system.

## 1. Introduction

The Brazilian health system comprises a two-tier system based on the principle of the promotion of universal health coverage for the population including the provision of integral healthcare without charge to patients accessing public and private healthcare facilities financed through the public sector (Sistema Único de Saúde, SUS), and the provision of healthcare financed through private disbursements from prepayments to health insurance companies or out-of-pocket expenditures for individuals with the capacity to pay. Approximately 75% of the Brazilian population depends on the provision of healthcare through the SUS. The major part of the remaining 25% of the population has health insurance. Thus, the right to healthcare has been guaranteed for the population since the inception of the current configuration of the national health system in 1988, maintained through government funding through tax revenues transferred to the SUS, being considered the largest national health system based on universal healthcare coverage worldwide [1,2]. However, universal healthcare coverage through the SUS presents limitations regarding infrastructure and human resources in vulnerable communities, settings with low accessibility, and low-income municipalities, in addition to budget fluctuations due to changes in tax revenue and inequalities in the distribution of financial resources at the national, state, and municipal levels [3,4,5].

The Brazilian national income tax represents one of the main sources of revenue to the SUS. However, health insurance beneficiaries obtain income tax deductions due to their access to private financed healthcare, thus averting the risk of double payment for healthcare [3,6,7]. Therefore, the utilization of healthcare by health insurance beneficiaries within the SUS generates expenditure to the public sector without the corresponding payment of income tax designed to finance the SUS, compromising its maintenance [8]. The income tax deductions for health insurance beneficiaries create incentives for health insurance companies by attracting additional consumers of health insurance whilst simultaneously reducing the government revenue to finance the SUS. Consequently, health insurance beneficiaries using healthcare within the SUS transfer healthcare disbursements to the government, generating additional financial benefits to health insurance companies by reducing their costs, particularly regarding high complexity procedures with higher costs [9]. In addition, income tax deductions may compromise the principle of taxation progressivity in the process of financing the national health system through government revenue, initially designed to ensure universal healthcare coverage predominantly based on the delivery of primary healthcare strategies for health promotion and disease prevention for lower income individuals throughout the country [1]. Therefore, the Brazilian government has established that health insurance companies with beneficiaries receiving publicly financed healthcare from the SUS should provide a reimbursement of the corresponding healthcare expenditure to the government. The requirement of reimbursement by health insurance companies was implemented in 1998 to discourage healthcare utilization within the SUS by health insurance beneficiaries [9,10].

Findings from a previous study showed that the motivation to access healthcare within the SUS, according to health insurance beneficiaries surveyed in the city of Sao Paulo during 2009, include the distance to the healthcare facilities covered by the health insurance, and the cost of transportation to access healthcare [11]. In addition, beneficiaries surveyed in the city of Rio de Janeiro during 2008 indicated that the use of healthcare within the SUS was compelled due to delays in authorization to perform healthcare procedures by health insurance companies [12]. Hence, an assurance of healthcare coverage that is free of charge to individuals within the SUS represents an additional economic incentive to the private health sector, encouraging health insurance companies to limit investments in infrastructure and human resources, in addition to imposing health insurance contracts with bureaucratic restrictions to access certain healthcare procedures [9].

Although the evidence in the literature indicates that charges for reimbursement to the SUS were concentrated in low- and medium-complexity healthcare procedures from 1999 to 2006, thus representing low financial burden to the SUS [11], a recent study showed that health insurance companies have been suing the government to circumvent reimbursement charges, indicating that the financial impact may be higher than the estimates in the literature [8]. Furthermore, the characteristics of the market for health insurance in Brazil show the substantial market power of companies due to increasing demand and a lack of competition in the sector [13]. In addition, complaints from health insurance beneficiaries against companies referring to restrictions in the access to healthcare procedures covered in health insurance contracts have been increasing during the last decades, albeit Brazilian laws established a list of mandatory procedures for health insurance coverage since 1998 [14].

Consequently, debates on the right to universal healthcare coverage in the country have been permeated by concerns on the economic sustainability of the SUS on the one hand, and the financial maintenance of health insurance companies on the other hand. In the policy arena, financial issues of the private sector generally tend to dominate agenda-setting processes, whereas the economic sustainability of the SUS usually comprises leading concern in the scientific approach of the Brazilian health system. However, there is a lack of recent studies exploring the evolution and financial burden to the government due to healthcare utilization patterns by health insurance beneficiaries within the SUS, with the majority of the evidence focused on potential impacts to health insurance companies, the analysis of healthcare utilization patterns during a limited timeframe, or the identification of specific diagnosis and healthcare procedures [15,16,17].

The investigation of government expenditure attributable to health insurance beneficiaries according to patterns of healthcare utilization contributes to the scientific debate by showing the limitations of previous evidence in the literature and proposing innovative approaches to foster the investigation of the subject. Furthermore, the results may support the definition of strategies toward the alignment of incentives to the public and private sectors in the policy arena. Therefore, the present study investigated patterns of healthcare utilization and public expenditure due to the use of public healthcare by beneficiaries of health insurance between 2003 and 2019. The hypotheses of the research include:
The utilization of healthcare within the SUS by health insurance beneficiaries was concentrated in high complexity procedures beyond emergency care, thus corresponding to substantial government expenditure;Healthcare utilization patterns by health insurance beneficiaries within the SUS throughout the period from 2003 to 2019 may be represented in clusters to support initiatives toward changes in discounts granted for health insurance beneficiaries to increase the tax revenue to finance public policies of health.

The remainder of the paper is organized as follows: Section 2 presents the materials and methods with subsections referring to the description of datasets, variables, and statistical analyses; Section 3 shows the results of the analyses; Section 4 presents the discussion; and finally, Section 5 presents our conclusions.

## 2. Materials and Methods

### 2.1. Datasets

The study was based on the quantitative analyses of individual-level datasets referring to the utilization of healthcare by beneficiaries of health insurance in the context of the SUS. Datasets including annual information on healthcare utilization by beneficiaries of health insurance from the National Agency of Supplementary Health (Agência Nacional de Saúde Suplementar, ANS) were organized into a single database comprising mixed cross-sectional data to allow for the identification of patterns of interest to inform public policies of health. These datasets are publicly available on the ANS platform, but have been anonymized to prevent the potential identification and/or disclosure of sensitive information of patients (https://dadosabertos.ans.gov.br/FTP/Base_de_dados/Microdados/dados_dbc/ressarcimento_ao_sus/ accessed on 19 November 2024). Additional information regarding the level of healthcare complexity was obtained from the Department of Informatics of the Brazilian Unified Health System (Departamento de Informática do SUS, DATASUS).

### 2.2. Variables

The variables included in the analyses of the study encompassed data in the following domains (Table 1):
Demographic and health characteristics of the patients: Sex, age, and health condition (based on the International Classification of Diseases, 10th. Revision, ICD-10);Healthcare utilization: Type of procedure, inpatient days, and public expenditure (disbursement per procedure or per patient per day, depending on the type of administrative record);Healthcare facility characteristics: Administrative level of the healthcare facility (according to the National Registry of Healthcare Facilities, Cadastro Nacional de Estabelecimentos de Saúde, CNES) and state.

The identification of the health condition of patients was based on the identification of groups of clinical diagnosis according to the ICD-10 chapters, and the assignment of the complexity level of the procedures was based on the DATASUS administrative codebook.

The following categories of clinical diagnosis were established according to the ICD-10 chapters: (1) infectious and parasitic diseases; (2) neoplasms; (3) diseases of blood, blood-forming organs, and disorders of immune mechanism; (4) endocrine, nutritional, and metabolic diseases; (5) mental, behavioral, and neurodevelopmental disorders; (6) diseases of the nervous system; (7) diseases of the eye and adnexa; (8) diseases of the ear and mastoid process; (9) diseases of the circulatory system; (10) diseases of the respiratory system; (11) diseases of the digestive system; (12) diseases of the skin and subcutaneous tissue; (13) diseases of the musculoskeletal system and connective tissue; (14) diseases of the genitourinary system; (15) pregnancy, childbirth, and the puerperium; (16) certain conditions originating in the perinatal period; (17) congenital malformations, deformations, and chromosomal abnormalities; (18) symptoms, signs, and abnormal clinical and laboratory findings; (19) injury, poisoning, and consequences of external causes; (20) external causes of morbidity and mortality; and (21) factors influencing health status and contact with health services.

The information on the characteristics of procedures corresponded to categorization according to the four levels of complexity listed in the System for Management of the Procedures, Medication, Orthosis, Prothesis, and Special Materials from DATASUS (Sistema de Gerenciamento da Tabela de Procedimentos, Medicamentos e Órteses, Próteses e Materiais Especiais do SUS, SIGTAP): high complexity, medium complexity, primary healthcare, and orthosis, prothesis, and special materials.

The organization of the dataset for analysis required the establishment of protocols for the data imputation of missing information referring to health conditions and procedures. The completion of registries of healthcare utilization without information on the health conditions of patients (n = 123,317) and registries of healthcare utilization without information on the procedures (n = 981) was based on cross-checking data: ICD-10 chapters were assigned according to the procedures prescribed for patients, and the complexity level of the procedure was assigned according to the combination of diagnosis and pattern of healthcare expenditure identified among other patients with a similar diagnosis. Finally, registries without information on diagnosis and procedure (n = 92) were considered missing cases.

The variable referring to the type of administrative record was based on the characteristic of the protocol for the payment of healthcare facilities adopted within the SUS: Authorization for Hospitalization (Autorização de Internação Hospitalar, AIH) or Authorization for High Complexity Ambulatory Procedures (Autorização de Procedimentos Ambulatoriais de Alta Complexidade, APAC).

The administrative level of the healthcare facility was defined according to the responsibility for the management of the resources within the healthcare unit: national, state, municipal, or private.

The public expenditure per procedure or disbursement per patient per day were updated to December 2019 based on deflators calculated using the National Consumer Price Index from the Brazilian Institute for Geography and Statistics (Índice de Preços ao Consumidor Amplo do Instituto Brasileiro de Geografia e Estatística, IPCA) and converted into international currency using the purchase power parity (PPP) factor for 2019, available at the World Bank platform [18,19], according to the Equation (1).
(1)$$E_{it}=\frac{VP_{it}\cdot ipca_{t}}{d_{it}\cdot ppp_{t}}$$
where *E_it_* = public expenditure with healthcare utilization by individual *i* in the period *t*; *ipca_t_* = deflator corresponding to the period *t*; *VP_it_* = value of the healthcare procedure for individual *i* in the period *i*; *d_it_* = inpatient days of the individual *i* in the period *t*; *ppp_t_* = PPP factor for 2019 in the period *t*.

### 2.3. Statistical Analyses

The study was based on the estimation of linear regression models for the analysis of factors associated with public expenditure attributable to healthcare utilization by health insurance beneficiaries within the SUS. The empirical strategy was founded on the identification of associations with the demographic and health characteristics of patients, the level of complexity of procedures performed on the patient, type of administrative record, ICD-10 chapter, and administrative level of the healthcare facility (Equation (2)).
(2)$$E_{it}=\beta_{1}\cdot X_{it}+\beta_{2}\cdot cid_{it}+\beta_{3}\cdot pcl_{it}+\beta_{4}\cdot f_{it}+\beta_{5}\cdot C_{it}+\varepsilon_{it}$$
where *E_it_* = public expenditure with healthcare utilization by individual *i* in the period *t*; *X_it_* = matrix of demographic characteristics of individual *i* in the period *t*; *cid_it_* = vector referring to the ICD-10 chapter of diagnosis for patient *i* in the period *t*; *pcl_it_* = vector describing the complexity level of the procedure conducted for patient *i* in the period *t*; *f_it_* = vector describing the managerial level of the healthcare facility attended by patient *i* in the period *t*; *C_it_* = matrix of control variables referring to the state, year, and interaction between state and year; *ε_it_* = error. Post-estimation tests referring to the variance inflation factor and covariance matrix were applied to avoid potential multicollinearity and covariance issues.

Finally, agglomerative hierarchical cluster analysis was performed based on the association of government expenditure in relation to demographic and health characteristics of health insurance beneficiaries accessing healthcare within the SUS. The hierarchical clustering was based on unsupervised machine learning techniques to support the identification of patterns of health insurance beneficiaries with a higher financial burden for the public sector throughout the period from 2003 to 2019.

Procedures adopted for the implementation of agglomerative hierarchical cluster analysis were based on three steps. First, principal component analysis (PCA) was applied to the data to streamline variables into three principal components, derived from the association matrices of healthcare expenditure across diverse groups of health insurance beneficiaries. Second, k-means clustering was performed to identify potential groups of individuals according to the similarities in patterns of public expenditure, demographic, and health characteristics. Third, the agglomerative hierarchical cluster technique was used to confirm patterns identified through k-means clustering, based on the first two principal components obtained in the PCA. The validation of clustering results was based on the qualitative assessment of cluster cohesion and separation, in addition to quantitative measures referring to the sum of squared errors (SSE) and silhouette analysis.

The regression analysis was performed using Stata software, version 17.0, adopting a significance level of *p* < 0.05, and the analysis of clusters according to the similarity of patterns was based on Python version 3.9 through the packages *scipy.cluster.hierarchy* and *sklearn.cluster*. Specifically, the package *scipy.cluster.hierarchy* was implemented to build hierarchical dendrograms, allowing for the differentiation of groups of health insurance beneficiaries, whilst the package *sklearn.cluster* was implemented for the identification of agglomerative clusters.

## 3. Results

The healthcare utilization within the SUS by health insurance beneficiaries encompassed approximately 4.3 million registries between 2003 and 2019, representing a low proportion of the health insurance coverage in the country (considered approximately 60 million individuals in 2019) [20]. Nevertheless, the financial resources required to provide procedures for health insurance beneficiaries within the SUS during the period of analysis corresponded to approximately 6.6 billion PPP dollars in 2019. There was a substantial increase in procedures performed for health insurance beneficiaries from 2012 to 2014 including higher public expenditure (Table 2).

The majority of health insurance beneficiaries using healthcare within the SUS were female throughout the period of analysis, and there was an increase in the proportion of adult and elderly individuals (55.5% to 58.7% of the procedures, respectively). The proportion of adults and elderly individuals increased from 2010 onwards, whereas the proportion of children, adolescents, and young adults presented a gradual decrease during the period (Table 3).

The utilization of high complexity procedures by health insurance beneficiaries can be seen to have increased substantially from 2012 onwards (pre-2012: 11.7% to 19.2%; post-2012: 30.6% to 38.1%). The majority of the procedures was performed in private healthcare facilities (44.5% to 64.1% of the procedures) or public healthcare facilities at the state level (18.3% to 31.0% of the procedures) (Table 4).

The coefficients of the regression model indicate that public expenditure with healthcare per capita per day for health insurance beneficiaries within the SUS was significantly associated with the demographic and health characteristics of the patients, healthcare complexity level, and the healthcare management characteristics (Table 5).

Factors linked to higher expenditure were being male patient aged ≥65 years old with treatment based on high complexity procedures, diagnosed with diseases of the eye and adnexa (ICD-10 Chapter 7), genitourinary system (ICD-10 Chapter 14), congenital malformations, deformations, and chromosomal abnormalities (ICD-10 Chapter 17). In contrast, lower expenditure was associated with female patients under 20 years of age, diagnosed with mental, behavioral, and neurodevelopmental disorders (ICD-10 Chapter 5), and diseases of the ear and mastoid process (ICD-10 Chapter 8). Health insurance beneficiaries receiving treatment in federal and municipal healthcare facilities presented a lower public expenditure (Table 5).

The higher financial burden was associated with the complexity level of procedures (β = 1.37; *p* < 0.001) and diseases of the genitourinary system (β = 0.89; *p* < 0.001) whereas lower financial impacts were identified for the treatment of mental, behavioral, and neurodevelopmental disorders (β = −1.00; *p* < 0.001) in healthcare facilities at the federal level (β = −0.21; *p* < 0.001).

The agglomerative hierarchical cluster analysis showed five patterns of healthcare utilization within the SUS according to the characteristics of the health insurance beneficiaries (Figure 1):
Cluster 1 (26.0% of registries, mean public expenditure = 686.51 dollars PPP in 2019): adult individuals 50–54 years old and elderly individuals ≥80 years old using medium complexity procedures related to infectious and parasitic diseases (ICD-10 Chapter 1), mental, behavioral, and neurodevelopmental disorders (ICD-10 Chapter 5), and diseases of the circulatory system (ICD-10 Chapter 9);Cluster 2 (25.1% of registries, mean public expenditure = 533.78 dollars PPP in 2019): children 10–14 years old and adult individuals 20–44 years old using medium complexity procedures related to diseases of the digestive system (ICD-10 Chapter 11), injury, poisoning, and consequences of external causes (ICD-10 Chapter 19), and factors influencing health status and linked to contact with health services (ICD-10 Chapter 21);Cluster 3 (26.1% of registries, mean public expenditure = 1876.23 dollars PPP in 2019): adult individuals 45–74 years old, usually female patients using high complexity procedures related to neoplasms (ICD-10 Chapter 2), diseases of the ear and mastoid process (ICD-10 Chapter 8), and genitourinary system (ICD-10 Chapter 14);Cluster 4 (8.2% of registries, mean public expenditure = 1317.99 dollars PPP in 2019): children less than 9 years old admitted for the treatment of diseases of the respiratory system (ICD-10 Chapter 8), predominantly beneficiaries of health insurance without hospitalization coverage;Cluster 5 (14.1% of registries, mean public expenditure = 288.56 dollars PPP in 2019): adolescents and adult individuals 15–39 years old, generally female patients using medium complexity procedures related to pregnancy, childbirth, and the puerperium (ICD-10 Chapter 15).

The distribution of cases within the agglomerative hierarchical cluster analysis identified proximity among health insurance beneficiaries in clusters 1, 2, and 5 (Figure 2).

## 4. Discussion

The findings of the study indicated low rates of healthcare utilization within the SUS by health insurance beneficiaries in comparison to the health insurance coverage rates in the population; however, the evolution of public expenditure for the payment of healthcare procedures during the period from 2003 to 2019 showed an increase in the financial burden assumed by the Brazilian government to the benefit of health insurance companies. In contrast to a previous study indicating an increase in public expenditure during 2011 and 2012 [8], the present results point to an increase in public expenditure per capita per day between 2012 and 2016. The differences in the trends of public expenditure for healthcare utilization among health insurance beneficiaries refer to the use of nominal public expenditure (instead of updated public expenditure), disregarding the effects of high inflation rates on monetary values in Brazil. The increase in public expenditure between 2012 and 2016 may be linked to the identification of the proper registry of high complexity ambulatory procedures in the datasets, although the regulatory agency for private healthcare officially recognized charges of ambulatory procedures starting in 2015 [21,22].

Patterns of healthcare utilization within the SUS and the evolution of public expenditure per capita per day among the health insurance beneficiaries in the present study showed divergences in relation to other previous studies [8,10,15,16,23] due to methodological differences. Certain studies focused on the analysis of data referring to hospitalizations, using data for limited periods and aggregated information at geographical level, while the current investigation included hospitalizations and high complexity ambulatory care [15,16,23]. Other studies failed to identify the application of deflators for updating monetary values [8,16], which distorts indicators due to the high inflation rates in Brazil.

In addition, the study highlighted higher public expenditure for the treatment of health insurance beneficiaries using high complexity healthcare procedures linked to diseases of the genitourinary system, particularly involving expensive protocols for the treatment of patients with chronic kidney disease (e.g., hemodialysis). The findings contradict previous evidence indicating the concentration of charges for reimbursement to the SUS due to utilization of low- and medium-complexity healthcare procedures from 1999 to 2006 [11]. The divergence between studies may be attributable to changes in the patterns of healthcare utilization post-2006, considering that the present study focused on the analysis of data between 2003 and 2019. The occurrence of demographic transition during the last decades in Brazil showed acceleration in population aging [1,4,13], representing direct impacts on patterns of healthcare demands.

In addition, signaling from lawsuits filed by health insurance companies against the Brazilian government, challenging reimbursement demands to the SUS, indicates a substantial burden attributable to healthcare utilization by health insurance beneficiaries [8]. The findings of the study are particularly noteworthy considering the rise in complaints from health insurance beneficiaries against health insurance companies due to restrictions in the access to healthcare procedures covered in health insurance contracts during the last decades [13].

Finally, the cluster analysis emphasized five patterns of healthcare utilization within the SUS throughout the period from 2003 to 2019, according to the characteristics of health insurance beneficiaries. Three clusters were marked by proximity in relation to emergency care: the first cluster including children with respiratory diseases without hospitalization coverage; the second cluster including adolescents and young adults with gastrointestinal diseases and issues linked to external causes; and the third including young and adult women using procedures related to pregnancy, childbirth, and the puerperium. However, the three clusters represented lower public expenditure in comparison to the remaining clusters, considering that the majority of healthcare utilization was attributable to low- and medium-complexity procedures associated with low public expenditure.

The two remaining clusters referred to expensive therapies linked to high complexity procedures related to the treatment of chronic diseases among adult individuals (e.g., neoplasms and hemodialysis), and medium complexity procedures for the treatment of cardiovascular diseases and senescence. Although previous studies have shown similarities in relation to the high concentration of healthcare utilization among women during pregnancy, childbirth, and the puerperium [15,16] (i.e., healthcare utilization among health insurance beneficiaries identified in cluster 5), the analysis proposed in the present study conveys additional evidence regarding other characteristics of patients and diagnosis linked to patterns of high and low public expenditure. In particular, the occurrence of high public expenditure due to high complexity procedures associated with aging showed the potential exploration of economic incentives by health insurance companies during the period of analysis [9], especially through rent-seeking strategies based on the delay of approval for expensive procedures prescribed to health insurance beneficiaries to foster the utilization of healthcare in public facilities.

The identification of healthcare utilization patterns by health insurance beneficiaries within the SUS allowed us to define the origin of the healthcare demand according to the patients’ diagnosis and costs of procedures, comprising evidence for decision-making processes in public policies of health in Brazil. Furthermore, considering the income tax incentives granted to health insurance beneficiaries, there is a lack of funding to the SUS for the coverage of healthcare demands arising from patients in the private health sector [24]. Therefore, strategies for improvement in the distribution of financial, structural, and human resources in the Brazilian national health system should encompass changes in tax incentives and equity in healthcare financing to address equity in the establishment of universal healthcare coverage for the population [25,26,27].

The study presents certain limitations, particularly due to the use of administrative datasets from the Brazilian regulatory agency for private healthcare. Considering that anonymized individual-level data publicly available on the platform of the National Agency of Supplementary Health prevented the identification of other characteristics of patients in the context of the analysis performed in the study, the investigation lacked an analysis of the patterns of healthcare utilization according to skin color/ethnicity, income level, and the follow-up of individuals throughout the period.

Furthermore, potential issues linked to data imputation techniques adopted in the study may have presented some failures. However, the protocol for data imputation regarding patient diagnosis and the complexity level of procedures was based on the interpolation of information referring to the patterns of healthcare utilization and costs extracted from the datasets to avoid potential biases arising from external criteria for data completion. Finally, it is important to highlight that the datasets were tested for consistency of the information including the adoption of robust techniques for statistical analyses with control variables referring to state, year, and cross effects between state and year to capture the potential effects at the local level, addressing a major part of the study limitations.

## 5. Conclusions

The first hypothesis of the research was confirmed through estimation of the regression model, which indicated a high effect of high complexity procedures on government expenditure with healthcare utilization among health insurance beneficiaries. Thus, the findings of the study highlight the existence of patterns of healthcare utilization by health insurance beneficiaries within the SUS that may compromise the economic sustainability of public funding within the Brazilian health system.

In addition, the second hypothesis of the research was confirmed through the identification of five clusters of health insurance beneficiaries using the SUS, showing lower public expenditure attributable to healthcare utilization among health insurance beneficiaries in emergency care. Low-expenditure clusters have usually been highlighted in previous studies to indicate the low economic impact of health insurance beneficiaries within the SUS; however, the high-expenditure clusters identified in the study contribute to the knowledge on the subject by presenting novel evidence in the field, considering the absence of previous explorations of the issue. These findings may be useful for the establishment or redesign of economic incentives for institutions in public and private sectors within the Brazilian health system.

Furthermore, additional features of the datasets may be incorporated into the further analyses of healthcare utilization patterns, allowing one to refine policy approaches to the economic sustainability of the SUS and to the financial maintenance of health insurance companies. Therefore, future explorations of the data may contribute to the identification of other groups of interest among health insurance beneficiaries.

In conclusion, the use of quantitative analyses based on the demographic and health characteristics of individuals allowed us to identify public expenditure associated with diverse patterns of healthcare utilization from 2003 to 2019. The evidence comprises useful information for the design of strategies addressing the financial constraints within the public sector in Brazil, particularly toward the promotion of equity in universal healthcare coverage for the population.

## Figures and Tables

**Figure 1 healthcare-12-02335-f001:**
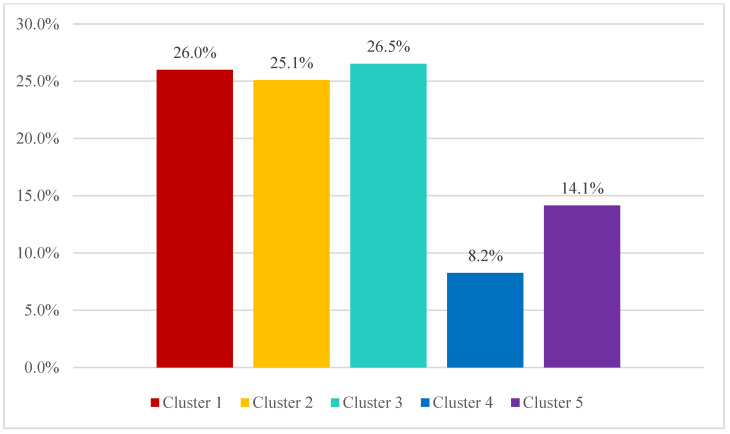
Distribution of healthcare procedures according to cluster. Brazil, 2003–2019.

**Figure 2 healthcare-12-02335-f002:**
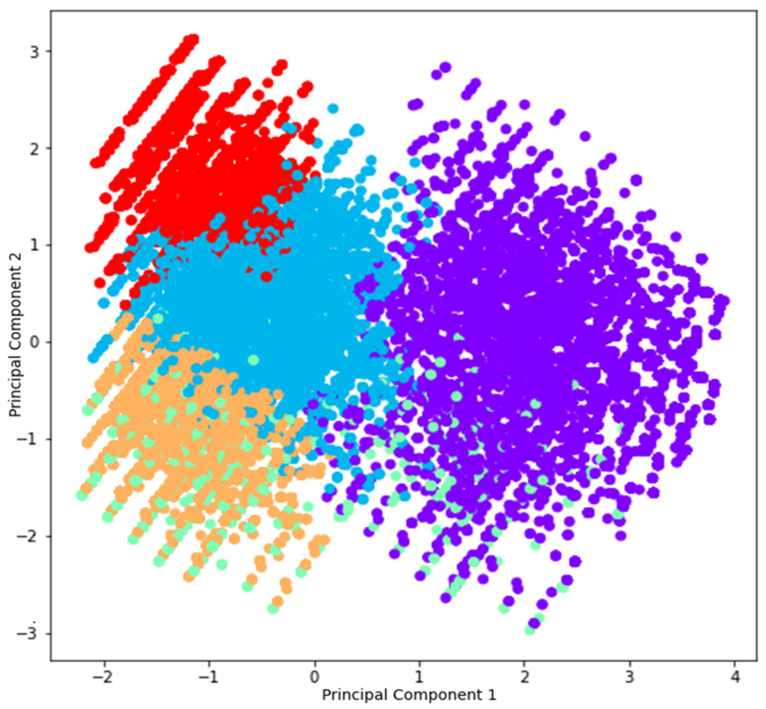
Characterization of clusters of public expenditure according to the utilization of healthcare by health insurance beneficiaries within the SUS. Brazil, 2003–2019. Obs.: 
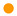
 Cluster 1; 
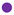
 Cluster 2; 
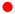
 Cluster 3; 
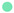
 Cluster 4; 
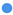
 Cluster 5. Graphic presenting the distribution of 1% cases (random sample of 43,306 cases) to allow for visualization of the clusters.

**Table 1 healthcare-12-02335-t001:** Characterization of variables and data sources. Brazil, 2003–2019.

Variable	N	μ	SD	Min	Max	Source
Sex	4,330,566	0.57	0.49	0	1	ANS
Age bracket	4,330,566	8.45	4.62	0	17	ANS
Public expenditure per day	4,330,567	763.42	2389.2	0	124,818.60	ANS
ICD-10 chapter	4,330,475	10.58	5.97	1	21	ANS
Complexity level of procedures	4,330,475	0.28	0.45	0	1	DATASUS
Type of administrative record	4,330,567	0.78	0.41	0	1	ANS
Administrative level of facility	4,207,102	3.15	1.02	1	4	ANS
State	4,321,029	34.09	7.56	11	53	ANS
Year	4,330,567	2012.23	4.46	2003	2019	ANS

Obs.: N = occurrences; μ = mean; SD = standard deviation; Min = minimum; Max = maximum.

**Table 2 healthcare-12-02335-t002:** Procedures and expenditure attributable to healthcare services for health insurance beneficiaries within the SUS, according to year. Brazil, 2003–2019.

Year	Procedures	Public Expenditure ($ PPP in 2019)			
		Value per Capita per Day		Total Value	
		μ	SE	μ	SE
2003	148,913	507.15	2.77	239	1.33
2004	165,747	462.14	2.59	243	1.31
2005	155,520	451.53	2.29	218	1.26
2006	158,982	458.68	2.48	239	1.35
2007	172,220	453.05	2.52	277	1.57
2008	169,229	399.26	3.11	238	1.76
2009	192,291	432.35	3.22	299	1.98
2010	243,355	425.18	2.93	372	2.26
2011	197,569	446.83	3.60	330	2.30
2012	393,259	973.36	4.65	688	3.08
2013	405,452	1033.66	5.29	734	3.50
2014	470,903	950.28	3.42	696	2.53
2015	321,572	922.93	4.86	485	2.19
2016	302,907	932.65	5.07	440	2.06
2017	285,039	941.36	5.38	412	2.01
2018	276,970	926.83	5.42	383	1.90
2019	270,639	874.29	5.23	347	1.69

Obs.: μ = mean; SE = standard error.

**Table 3 healthcare-12-02335-t003:** Characterization of health insurance beneficiaries using healthcare within the SUS. Brazil, 2003–2019.

Year	Sex				Age Bracket									
	Male		Female		<10 Years Old		10–14 Years Old		15–39 Years Old		40–74 Years Old		≥75 Years Old	
	%	95%CI	%	95%CI	%	95%CI	%	95%CI	%	95%CI	%	95%CI	%	95%CI
2003	41.8	41.5; 42	58.2	58; 58.5	13.3	13.1; 13.5	2.4	2.3; 2.5	44.4	44.1; 44.6	34.6	34.4; 34.9	5.3	5.2; 5.4
2004	42.4	42.1; 42.6	57.6	57.4; 57.9	12.5	12.4; 12.7	2.3	2.2; 2.4	44.3	44.1; 44.5	35.1	34.8; 35.3	5.8	5.7; 5.9
2005	42.2	42.0; 42.5	57.8	57.5; 58.0	13.0	12.8; 13.2	2.5	2.4; 2.6	43.2	43; 43.4	35.9	35.7; 36.1	5.4	5.3; 5.5
2006	44.5	44.2; 44.7	55.5	55.3; 55.8	13.9	13.7; 14.1	2.6	2.5; 2.7	43.7	43.5; 43.9	34.6	34.4; 34.8	5.2	5.1; 5.3
2007	43.1	42.8; 43.3	56.9	56.7; 57.2	15.8	15.7; 16.0	2.7	2.6; 2.8	44.4	44.1; 44.6	32.4	32.2; 32.6	4.7	4.6; 4.8
2008	42.3	42.1; 42.6	57.7	57.4; 57.9	16.2	16.1; 16.4	2.8	2.7; 2.9	45.3	45.1; 45.6	31.0	30.8; 31.2	4.6	4.5; 4.7
2009	42.5	42.3; 42.7	57.5	57.3; 57.7	15.6	15.5; 15.8	2.9	2.9; 3.0	45.6	45.3; 45.8	30.9	30.7; 31.1	5.0	4.9; 5.1
2010	42.6	42.4; 42.8	57.4	57.2; 57.6	16.3	16.2; 16.4	3.0	3.0; 3.1	45.3	45.1; 45.5	30.3	30.1; 30.4	5.1	5.0; 5.2
2011	43.5	43.3; 43.7	56.5	56.3; 56.7	13.4	13.3; 13.6	3.2	3.1; 3.2	43.7	43.5; 43.9	33.6	33.4; 33.8	6.1	6.0; 6.2
2012	42.1	41.9; 42.2	57.9	57.8; 58.1	11.4	11.3; 11.5	2.5	2.4; 2.5	38.1	38.0; 38.3	39.6	39.4; 39.7	8.3	8.3; 8.4
2013	41.3	41.1; 41.4	58.7	58.6; 58.9	11.6	11.5; 11.7	2.5	2.4; 2.5	38.1	38.0; 38.3	39.3	39.1; 39.4	8.5	8.4; 8.6
2014	41.9	41.8; 42.1	58.1	57.9; 58.2	9.6	9.5; 9.7	2.4	2.3; 2.4	32.3	32.2; 32.4	44.6	44.5; 44.8	11.1	11; 11.2
2015	42.9	42.7; 43.1	57.1	56.9; 57.3	11.1	11.0; 11.2	2.5	2.4; 2.5	34.9	34.7; 35.0	40.9	40.7; 41.1	10.7	10.6; 10.8
2016	43.0	42.8; 43.1	57.0	56.9; 57.2	11.3	11.1; 11.4	2.5	2.4; 2.5	34.1	33.9; 34.3	42.2	42.0; 42.4	10.0	9.9; 10.1
2017	43.1	42.9; 43.3	56.9	56.7; 57.1	11.1	10.9; 11.2	2.5	2.5; 2.6	33.4	33.2; 33.5	42.8	42.6; 43.0	10.2	10.1; 10.4
2018	43.4	43.2; 43.6	56.6	56.4; 56.8	11.3	11.2; 11.5	2.6	2.5; 2.6	32.8	32.6; 33	43.0	42.8; 43.2	10.3	10.2; 10.4
2019	43.3	43.1; 43.5	56.7	56.5; 56.9	11.5	11.3; 11.6	2.6	2.5; 2.6	32.6	32.4; 32.8	43.2	43.1; 43.4	10.1	10.0; 10.3

Obs.: 95%CI = 95% confidence interval.

**Table 4 healthcare-12-02335-t004:** Characterization of the complexity level of procedures and the administrative level of healthcare facilities. Brazil, 2003–2019.

Year	Complexity Level				Administrative Level of Healthcare Facility							
	Low and Medium		High		Federal		State		Municipal		Private	
	%	95%CI	%	95%CI	%	95%CI	%	95%CI	%	95%CI	%	95%CI
2003	81.2	81; 81.4	18.8	18.6; 19.0	6.1	6.0; 6.3	18.3	18.1; 18.5	11.4	11.2; 11.6	64.1	63.9; 64.4
2004	81.4	81.3; 81.6	18.6	18.4; 18.7	6.4	6.2; 6.5	25.5	25.3; 25.7	16.1	15.9; 16.3	52.0	51.7; 52.3
2005	80.8	80.6; 81.0	19.2	19.0; 19.4	5.1	5.0; 5.3	24.1	23.8; 24.3	9.3	9.2; 9.5	61.5	61.2; 61.7
2006	81.1	80.9; 81.3	18.9	18.7; 19.1	5.9	5.8; 6.0	25.6	25.4; 25.8	14.2	14.1; 14.4	54.3	54.0; 54.5
2007	81.9	81.7; 82.1	18.1	17.9; 18.3	6.6	6.5; 6.7	27.8	27.6; 28.0	16.0	15.8; 16.2	49.6	49.4; 49.8
2008	88.3	88.1; 88.4	11.7	11.6; 11.9	6.1	6.0; 6.2	26.7	26.5; 26.9	16.5	16.4; 16.7	50.6	50.4; 50.9
2009	86.9	86.7; 87.0	13.1	13.0; 13.3	6.6	6.5; 6.7	27.7	27.5; 27.9	17.0	16.8; 17.1	48.7	48.5; 48.9
2010	86.6	86.4; 86.7	13.4	13.3; 13.6	6.4	6.3; 6.5	29.0	28.8; 29.1	17.7	17.5; 17.8	47.0	46.8; 47.2
2011	84.3	84.1; 84.4	15.7	15.6; 15.9	6.8	6.7; 7.0	31.0	30.8; 31.2	17.6	17.4; 17.8	44.5	44.3; 44.7
2012	69.4	69.3; 69.6	30.6	30.4; 30.7	6.7	6.6; 6.7	24.5	24.4; 24.7	16.4	16.3; 16.5	52.4	52.3; 52.6
2013	68.4	68.3; 68.6	31.6	31.4; 31.7	7.2	7.1; 7.3	24.9	24.8; 25.0	14.5	14.4; 14.6	53.3	53.2; 53.5
2014	61.9	61.8; 62.0	38.1	38.0; 38.2	6.1	6.0; 6.2	23.2	23.1; 23.3	11.9	11.8; 12.0	58.8	58.7; 59.0
2015	68.6	68.4; 68.7	31.4	31.3; 31.6	7.0	6.9; 7.1	24.8	24.6; 24.9	14.3	14.2; 14.5	53.9	53.7; 54.1
2016	65.0	64.8; 65.2	35.0	34.8; 35.2	7.2	7.2; 7.3	24.7	24.6; 24.9	14.0	13.9; 14.1	54.0	53.8; 54.2
2017	64.6	64.4; 64.8	35.4	35.2; 35.6	7.4	7.3; 7.5	24.6	24.5; 24.8	13.7	13.6; 13.8	54.2	54.1; 54.4
2018	64.2	64.0; 64.4	35.8	35.6; 36.0	7.5	7.4; 7.6	23.9	23.8; 24.1	13.9	13.8; 14.1	54.7	54.5; 54.9
2019	63.7	63.5; 63.9	36.3	36.1; 36.5	7.5	7.4; 7.6	23.9	23.8; 24.1	14.3	14.2; 14.5	54.3	54.1; 54.5

Obs.: 95%CI = 95% confidence interval.

**Table 5 healthcare-12-02335-t005:** Coefficients of the linear regression model for public expenditure per capita per day. Brazil, 2003–2019.

Variable		β	SE	Sig.	95%CI
Sex	(fem = 1)	−0.07	0.00	***	−0.07; −0.07
Age bracket					
<1 year	(yes = 1)	−0.26	0.00	***	−0.27; −0.25
1–4 years	(yes = 1)	−0.17	0.00	***	−0.17; −0.16
5–9 years	(yes = 1)	−0.13	0.00	***	−0.14; −0.12
10–14 years	(yes = 1)	−0.16	0.00	***	−0.17; −0.16
15–19 years	(yes = 1)	−0.12	0.00	***	−0.12; −0.11
20–24 years	(yes = 1)	−0.10	0.00	***	−0.10; −0.09
25–29 years	(yes = 1)	−0.08	0.00	***	−0.09; −0.07
30–34 years	(yes = 1)	−0.09	0.00	***	−0.09; −0.08
35–39 years	(yes = 1)	−0.10	0.00	***	−0.10; −0.09
40–44 years	(yes = 1)	−0.11	0.00	***	−0.11; −0.10
45–49 years	(yes = 1)	−0.11	0.00	***	−0.11; −0.10
50–54 years	(yes = 1)	−0.09	0.00	***	−0.09; −0.08
55–59 years	(yes = 1)	−0.06	0.00	***	−0.07; −0.06
60–64 years	(yes = 1)	0.00	0.00		0.00; 0.01
65–69 years	(yes = 1)	0.02	0.00	***	0.01; 0.02
70–74 years	(yes = 1)	0.02	0.00	***	0.02; 0.03
75–79 years	(yes = 1)	0.03	0.00	***	0.02; 0.04
High complexity procedure	(yes = 1)	1.37	0.00	***	1.37; 1.38
Chapter ICD-10					
Chapter 2	(yes = 1)	0.56	0.00	***	0.55; 0.56
Chapter 3	(yes = 1)	−0.18	0.01	***	−0.20; −0.17
Chapter 4	(yes = 1)	0.12	0.00	***	0.11; 0.13
Chapter 5	(yes = 1)	−1.00	0.00	***	−1.00; −0.99
Chapter 6	(yes = 1)	−0.04	0.00	***	−0.05; −0.04
Chapter 7	(yes = 1)	0.74	0.00	***	0.74; 0.75
Chapter 8	(yes = 1)	−0.22	0.00	***	−0.22; −0.21
Chapter 9	(yes = 1)	0.43	0.00	***	0.42; 0.43
Chapter 10	(yes = 1)	0.38	0.00	***	0.38; 0.39
Chapter 11	(yes = 1)	0.58	0.00	***	0.57; 0.59
Chapter 12	(yes = 1)	0.17	0.00	***	0.16; 0.17
Chapter 13	(yes = 1)	0.36	0.00	***	0.35; 0.37
Chapter 14	(yes = 1)	0.89	0.00	***	0.88; 0.89
Chapter 15	(yes = 1)	0.67	0.00	***	0.67; 0.68
Chapter 16	(yes = 1)	0.49	0.01	***	0.46; 0.51
Chapter 17	(yes = 1)	0.76	0.00	***	0.75; 0.77
Chapter 18	(yes = 1)	−0.06	0.00	***	−0.07; −0.05
Chapter 19	(yes = 1)	0.36	0.00	***	0.35; 0.37
Chapter 20	(yes = 1)	−0.01	0.02		−0.04; 0.02
Chapter 21	(yes = 1)	0.73	0.00	***	0.73; 0.74
Administrative level					
Federal	(yes = 1)	−0.21	0.00	***	−0.21; −0.20
Municipal	(yes = 1)	−0.13	0.00	***	−0.13; −0.13

Obs.: β = coefficient; SE = standard error; Sig. = significance; 95%CI = 95% confidence interval; ICD-10 = International Classification of Diseases, 10th. Revision. *** *p* < 0.001.

## Data Availability

Datasets in the present study are publicly available on the platform of the ANS (https://dadosabertos.ans.gov.br/FTP/Base_de_dados/Microdados/dados_dbc/ressarcimento_ao_sus/ accessed on 19 November 2024).

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
