# Peer review of "Determinants of Government Expenditures with Health Insurance Beneficiaries in the Brazilian Health System"

_healthcare, 2024, doi:10.3390/healthcare12232335_

Round 1
Reviewer 1 Report
Comments and Suggestions for Authors
We dont' really see the point that the authors try to deal with.
The paper demonstrate the existence of patterns of healthcare utilization, and public expenditures, among beneficiaries of health insurance. OK, there are clusters of utilisation of care; this is expected: people with needs of healthcare are specific. We don't see why do the authors deliver this result, and why it is important. In other words: the problematization (introduction) is very poorly formulated.
There is no logical connexion with the fact that "it may compromise the sustainability of the Brazilian healthcare system". The "what-this-paper-adds" is totally unclear.
Reviewer 2 Report
Comments and Suggestions for Authors
This paper focuses on investigating healthcare utilization and public expenditures in the context of the Brazilian health system.
- The title and the content of the study are not quite consistent. The former stresses studying patterns of healthcare utilization, while the content is actually an analysis of the factors influencing health care expenditures. It is suggested that the title be adjusted.
- The fact that a few studies investigate this issue does not necessarily mean that the issue has academic value and relevance for research. The author should seriously discuss the academic value and practical significance of studying the patterns of healthcare utilization and health care expenditure of the Brazilian health system in the introduction part of the paper.
- The paper should discuss in the introduction section what the specific contributions of this paper are in relation to the existing literature;
- It is recommended that author add a paragraph in the introduction to describe the structure of the paper.
- The data in the paper involves individual and time dimensions, and the error terms in the model (2) should have individual and time subscripts;
- Authors should specify whether the dataset used in the paper is panel data or mixed cross-section data;
- The model (2) is a panel data model, so why not consider individual fixed effects and time fixed effects?
- From Table 5, the regression model has too many variables, and the estimation results are easily disturbed by the covariance problem. It is recommended that the authors use principal component analysis to streamline the variables.
Authors should improve English expression as appropriate.
Reviewer 3 Report
Comments and Suggestions for Authors
The article is dedicated to investigating the specific features of Brazil's healthcare system and its patients. The authors define some patterns of healthcare utilization within this health system. Without a doubt, the paper has pluses. It is well structured with good mathematical apparatus, including a huge base of statistical information for processing. It could be a complex theoretical contribution to research geared toward healthcare. In my opinion, the manuscript is scientifically sound and gives appropriate and understandable results for its reader, but there are some minuses of the paper.
Here are some other deficiencies that need to be further improved:
1) I recommend adding keywords (line 31), not the same as in the title of the paper.
2) Line 34-38 fully repeat the sentences in the Abstract of the paper (lines 8-12).
3) Line 234. Enrich the text with an explanation of the approach of conducting cluster analysis, the program solution used for it, and the validation of results.
4) Figure 2. Why are there minus values in the diagram for Healthcare utilization and Public expenses?
5) The Conclusions chapter is too short. Expand it with statements about the rare patterns of healthcare utilization, the proven and unproven hypotheses of the research, and the novelty of the research.
Round 2
Reviewer 1 Report
Comments and Suggestions for Authors
The manuscript has improved
I still have some concerns.
1/ I don't understand hypothesis 2.
"Healthcare utilization patterns by health insurance beneficiaries within the SUS 120 throughout the period from 2003 to 2019 may be defined through agglomerative 121 hierarchical cluster analysis to support changes in public policies of health."
Please, rephrase for having something less technical. What is the meaning (/link) of this assumption for the sustanalibility. This is not clear
2/ I don't understand this sentence:
"In particular, the occurrence of high public expenditures due to high complexity procedures associated with ageing howed potential exploration of economic incentives by health insurance companies during the period of analysis"
3/ I dont' understand the conclusion:
the high-expenditure clusters identified in the study contribute to the knowledge on the subject by presenting novel evidence in the field. The findings may be useful for establishment or redesign of economic incentives for institutions in public and private sectors within the Brazilian health system
1+2+3: I don't understand why the fact of a cluster of high consummers would necessary create unsustainability ; this is the very nature of a public health insurance system to insure high risk persons.
Author Response
Thank you for your time and suggestions, please see the attachment.

Reviewer 2 Report
Comments and Suggestions for Authors
The paper has been revised.
Reviewer 3 Report
Comments and Suggestions for Authors
Add more keywords. Three is too few.
Author Response

(The authors gave the same response as above.)
